# Effects of High Intensity Ultrasound on Physiochemical and Structural Properties of Goat Milk β-Lactoglobulin

**DOI:** 10.3390/molecules25163637

**Published:** 2020-08-10

**Authors:** Xinhui Zhou, Cuina Wang, Xiaomeng Sun, Zixuan Zhao, Mingruo Guo

**Affiliations:** 1Key Laboratory of Dairy Science, Northeast Agricultural University, Harbin 150030, China; zhouxh15@mails.jlu.edu.cn (X.Z.); wangcuina@jlu.edu.cn (C.W.); sunxm@neau.edu.cn (X.S.); zhaozixuan1006@126.com (Z.Z.); 2Department of Food Science, College of Food Science and Engineering, Jilin University, Changchun 130062, China; 3Department of Nutrition and Food Sciences, College of Agriculture and Life Sciences, University of Vermont, Burlington, VT 05405, USA

**Keywords:** goat milk β-lactoglobulin, high intensity ultrasound, physiochemical property, structure

## Abstract

This study aimed to compare the effects of high intensity ultrasound (HIU) applied at various amplitudes (20~40%) and for different durations (1~10 min) on the physiochemical and structural properties of goat milk β-lactoglobulin. No significant change was observed in the protein electrophoretic patterns by sodium dodecyl sulfate polyacrylamide gel electrophoresis (SDS-PAGE). Deconvolution and second derivative of the Fourier transform infrared spectra (FTIR) showed that the percentage of β-sheet of goat milk β-lactoglobulin was significantly decreased while those of α-helix and random coils increased after HIU treatment The surface hydrophobicity index and intrinsic fluorescence intensity of samples was enhanced and increased with increasing HIU amplitude or time. Differential scanning calorimetry (DSC) results exhibited that HIU treatments improved the thermal stability of goat milk β-lactoglobulin. Transmission electron microscopy (TEM) of samples showed that the goat milk β-lactoglobulin microstructure had changed and it contained larger aggregates when compared with the untreated goat milk β-lactoglobulin sample. Data suggested that HIU treatments resulted in secondary and tertiary structural changes of goat milk β-lactoglobulin and improved its thermal stability.

## 1. Introduction

β-Lactoglobulin is a major whey protein in ruminant milk (~50%, *w/w*). It mainly exists in the form of a dimer at its physiological pH [1]. β-lactoglobulin monomer is a compact molecule with a molecular weight at 18.3 kDa. It contains five cysteine residues with four of them forming two disulfide bonds and one free thiol group [2]. Studies showed that β-lactoglobulin molecule contains nine β-sheets and one α-helix [3]. For bovine milk, this protein has a shape of a “calyx”, which was composed of eight β-sheets and the ninth β-sheet forms a part of the β-lactoglobulin dimer interface [4,5]. However, the general structure of β-lactoglobulin from different sources may have different specific amino acid sequence and structure. Goat milk β-lactoglobulin is similar to bovine milk β-lactoglobulin [6], but there are eight amino acid substitutions at *N*-terminal Leu, Asp 53, Asp 64, Val 118, Asp 130, Ser 150, Glu 158, and Ile 162 in bovine milk β-lactoglobulin (database accession number, PDB 3BLG) compared with *N*-terminal Ile, Asn 53, Gly 64, Ala 118, Lys 130, Ala 150, Gly 158, and Val 162 in goat milk β-lactoglobulin (PDB 4OMX). High similarity in sequence between bovine and goat milk β-lactoglobulin suggests that both proteins should have similar physiological property and function [7]. There are differences in isoelectric point, stability of native dimers, thermal denaturation temperature, denaturation reaction rate, susceptibility to chemicals, and affinity for fatty acids between the two proteins [6,8].

Bovine milk β-lactoglobulin is by far the most often studied whey protein. There is limited information about the structural properties of goat milk β-lactoglobulin. Ultrasound technology is based on mechanical waves at a frequency above the threshold of human hearing (>16 kHz). Typically, intensity and frequency of ultrasound at range from 10~1000 W cm^−2^ and 16~100 kHz is considered as high intensity ultrasound (HIU) [9]. Nowadays, HIU has become a commonly used non-thermal technique in food science [10]. The application of non-thermal techniques has potential to modify the functional properties of proteins, thus recently, this technique has become a great area of interest for researchers. The mechanisms of HIU treatment on food materials are related to cavitation, heating, dynamic agitation, shear stress, and turbulence [11,12]. Certain studies have investigated the changes in molecular structure after HIU treatment, which induced alterations in free sulfhydryl groups, particle sizes, surface hydrophobicity, and secondary structures of proteins [11,13,14]. Several authors have speculated that the impact of ultrasound on the chemical and physical changes of protein, mainly due to higher shear forces, resulted in the breakdown of polypeptide chains, thus, disruption of intermolecular disulfide and non-covalent hydrophobic interactions [12,15,16]. HIU can modify the secondary and tertiary structures of proteins, which lead to the changes in the functional properties such as emulsifying activity, gelling properties, and foaming capacity of proteins [17,18,19,20].

Information about the effects of HIU treatment on the physiochemical and structural properties of goat milk β-lactoglobulin is very limited. This study aimed to investigate HIU treatment induced changes in physiochemical and structural properties of goat milk β-lactoglobulin by using fluorescence spectra, Fourier transform infrared spectra (FTIR), native and sodium dodecyl sulfate (SDS) polyacrylamide gel electrophoresis (PAGE), differential scanning calorimetry (DSC), and transmission electron microscopy (TEM).

## 2. Results and Discussion

### 2.1. Comparison between Native Bovine Milk β-Lactoglobulin and Goat Milk β-Lactoglobulin Using Reversed Phase High Performance Liquid Chromatography (RP-HPLC) and Sodium Dodecyl Sulfate (SDS) Polyacrylamide Gel Electrophoresis (PAGE) 

Differences in retention time of RP-HPLC between bovine milk β-lactoglobulin and isolated goat milk β-lactoglobulin were shown in Figure 1. Goat milk β-lactoglobulin samples showed considerable differences in the shape of elution peaks and retention time from bovine milk β-lactoglobulin. The retention times of the main peaks for goat milk β-lactoglobulin were at 19.060 and 19.337 min (Figure 1b) while those for bovine milk β-lactoglobulin were 19.773 and 19.983 min (representing bovine milk β-lactoglobulin A and B variants, Figure 1a). Different retention times between bovine and goat milk β-lactoglobulin may be due to the different molecule polarity. Compared with bovine milk β-lactoglobulin, eight different amino acids were found in goat milk β-lactoglobulin molecule primary structure, which may lead to a slightly different polarity. SDS-PAGE (Figure 1c) indicated that the molecular weight of bovine β-lactoglobulin and goat β-lactoglobulin was very close to that reported in [6].

### 2.2. Effects of HIU Treatment on Surface Hydrophobicity Index of Goat Milk β-Lactoglobulin

Surface hydrophobicity index (H_0_) is an indicator of hydrophobicity groups that are exposed on the surface of the protein molecule [21]. 8-Anilino-1-naphtalensulfonic acid (ANS) addition could help enhance the fluorescence strength of the protein’s surface hydrophobic pockets [22]. As a lipocalin protein, goat milk β-lactoglobulin is known to have surface hydrophobic pockets or clefts [3]. Surface hydrophobicity indexes of the samples were measured using the ANS method. Compared with the control, the H_0_ values of the HIU treated samples increased (*p* < 0.05) (Figure 2). Similar results for HIU induced increase in surface hydrophobicity for casein, soy protein isolate, and pea proteins were reported, respectively [13,14,15]. Ultrasound treatment may cause unfolding of the β-lactoglobulin molecules, which could expose more hydrophobic groups and regions initially bound inside.

H_0_ of all samples increased with an ultrasound time increase from 0 to 10 min regardless of amplitude. No significant difference between samples treated for seven and 10 min at all three amplitude levels was observed (*p* > 0.05). However, previous study showed that surface hydrophobicity of whey protein was decreased when HIU treating time was prolonged more than 5 min [17]. The H_0_ of the ultrasound treated samples increased due to protein unfolding or decreased due to protein aggregation, which was closely related to the protein type and composition as well as the ultrasound conditions [23].

### 2.3. Effects of HIU Treatment on Intrinsic Fluorescence of Goat Milk β-Lactoglobulin

The intrinsic fluorescence of aromatic amino acids in proteins has been used as a method for detecting conformational changes [24]. There are three amino acid residuals (tryptophan, Trp; tyrosine, Tyr; phenylalanine, Phe) that can emit fluorescence [16]. The excitation wavelengths of Trp residues range from 285 to 305 nm, and the Tyr residues were at 275 nm. When excited at 295 nm, only Trp residues can emit intrinsic fluorescence. Intrinsic fluorescence spectra at an excitation wavelength of 295 nm are shown in Figure 3. Compared with the control, no obvious changes in the shape of fluorescence spectra were observed for samples after HIU treatment. However, the emission maximum (λ_max_) of goat milk β-lactoglobulin shifted from 328 nm to 330 nm by HIU treatment regardless of ultrasonic amplitude and duration. The red shift of fluorescence emission wavelength suggested that more fluorescent amino acids were exposed into a polarity environment, which resulted in the change of conformation of protein [16]. The fluorescence intensity of the samples was enhanced and increased with increasing amplitude or time. The data indicated that sonication induced the exposure of more Trp side chains from the inside of the molecules. Partial unfolding of the protein molecular as a result of the HIU treatment might decrease the internal quenching, which contributed to the increase of fluorescence intensity [25]. Similar results were observed by a previous study [26].

### 2.4. Effects of HIU Treatment on the Secondary Structure of Goat Milk β-Lactoglobulin

Amide I band (1600~1700 cm^−1^) of the FTIR spectra of the β-lactoglobulin with different HIU amplitudes and treatment times were used to quantify the changes in secondary structures of β-lactoglobulin. Sonication caused a reduction in the percentage of β-sheet, while there was an increase in that of α-helix and random coil for the goat milk β-lactoglobulin samples (*p* < 0.05) (Figure 4). This phenomenon indicated that the β-lactoglobulin was more disordered, which was presumably attributed to the transformation of the β-sheet into random coils. The interior unfolding process of goat milk β-lactoglobulin molecules may also increase random coil structures, and a similar result was reported in previous studies [26,27,28]. Goat milk β-lactoglobulin is known as a β-sheet protein, and the reduction of the β-sheet induced by HIU may be responsible for the changes in intrinsic fluorescence. Bovine milk β-lactoglobulin has two Trp residues (Trp-19 and Trp-61) and Trp-19 located at the bottom of hydrophobic calyx is mainly responsible for the intrinsic fluorescence [24]. Reduction of β-sheet in goat milk β-lactoglobulin after HIU treatment may involve partial collapse of the calyx. The hydrophobic regions of native goat milk β-lactoglobulin were buried within the calyx, while the cavitation phenomenon induced by ultrasonic treatment could expose some of the hydrophobic regions to the surface, which could increase the value of H_0_ and intrinsic fluorescence intensity.

### 2.5. Effects of HIU Treatment on Molecular Weight of Goat Milk β-Lactoglobulin

Effects of HIU treatment on the protein profile of goat milk β-lactoglobulin were analyzed using SDS-PAGE (reducing and non-reducing) and native-PAGE. SDS disrupts the non-covalent bonds while dithiothreitol (DTT) further dissociates the disulfide bonds under reducing conditions [29]. Non-covalent hydrophobic interactions in protein could be evaluated by native-PAGE and non-reducing SDS-PAGE. Untreated sample showed two remarkable bands located at approximately 18.3 kDa and 36.71 kDa (Figure 5a–c), which represents the monomer and dimer form of goat milk β-lactoglobulin, respectively. The relative band intensity of 36.71 kDa is markedly increased in native PAGE (Figure 5c) than that in SDS-PAGE (Figure 5a,b). The dimer goat β-lactoglobulin was disrupted by SDS in SDS-PAGE. Compared with native β-lactoglobulin, there were no changes in the electrophoretic profiles of all experimental samples, which indicates no intermolecular covalent (disulfide) bonds formed during HIU treatment. This was in agreement with the results for bovine β-lactoglobulin, ovalbumin, and black bean protein isolate, which were treated by HIU [16,30,31]. However, the molecular weight of bovine milk β-lactoglobulin has been reported to be increased after HIU treatment [26]. This difference may be due to the uncontrolled sonication temperature used in previous studies. Acoustic cavitation, which is the nucleation, growth and collapse of gaseous bubbles within a fluid, was formed during ultrasound treatment. Mechanism of HIU were attributed to the shear force during the collapse of tiny bubbles. However, high temperature (up to 5000 °C) was accompanied by this process, which could significantly change the protein (thermal-sensitive) structure and molecular aggregation [28].

### 2.6. Effects of HIU Treatment on Thermal Properties of Goat Milk β-Lactoglobulin

Thermal properties of goat milk β-lactoglobulin treated by HIU were measured and the results are shown in Figure 6. Native β-lactoglobulin exhibited a single endothermic transition at the peak temperature (T_peak_) of 75.1 °C, which was similar to the findings by Shin et al. [32]. Ultrasound treatment raised the T_peak_ of all treated samples and the value of T_peak_ increased with increasing amplitude and time. The results suggest that the thermal stability of the protein was improved by HIU treatment and the stability was further improved by increasing amplitude and duration. HIU treatment induced a certain degree of molecular unfolding of the goat milk β-lactoglobulin, which disordered the stable compact fold of native goat milk β-lactoglobulin. More energy may be required to denature the unfolding structure by dissociation of intramolecular bonds such as covalent bonds.

### 2.7. Effects of HIU Treatment on Microstructure of Goat Milk β-Lactoglobulin

For TEM images (Figure 7), the white areas represented molecules of goat milk β-lactoglobulin. Native goat milk β-lactoglobulin (Control) has a spherical shape with a diameter of about 3.6 nm. After HIU treatment, samples showed larger irregular microscopic particle clusters >10 nm than the control. Goat milk β-lactoglobulin treated by HIU may cause aggregation (protein–protein interactions). According to the previous report, hydrophobic interactions, electrostatic bonds, and van der Waals interactions between molecules may be involved in this process.

## 3. Materials and Methods

### 3.1. Materials

Bovine β-lactoglobulin standard (≥90% purity, lyophilized powder) and 8-anilino-1-naphtalensulfonic acid (ANS) were obtained from Sigma-Aldrich (City of Saint Louis, MO, USA). Tris base (ultra-pure) and sodium dodecyl sulfate (>99%) were purchased from Solarbio Company (Beijing, China). High Performance Liquid Chromatography (HPLC)-grade solvent acetonitrile was purchased from Fisher Scientific (Fair Lawn, NJ, USA). All other chemicals were obtained from Beijing Chemical Works (Beijing, China). The water used in this study was filtered by Millipore Milli-Q water purification system (Millipore Corp., Milford, MA, USA).

### 3.2. Preparation of Goat Milk β-Lactoglobulin

Goat milk β-lactoglobulin was isolated from raw goat milk according to a previous method with some modifications [33]. Briefly, raw goat milk was heated to 40 °C, and then added with Na_2_SO_4_ (20 g/100 mL) and stirred at 150 rpm. After the dissolution of salt, the mixture was filtered through Whatman No. 4 filter paper (GE Healthcare, Boston, MA, USA). The precipitate was discarded. Filtrate (F1) was collected and adjusted to pH 2.0 using 5 M HCl. Then the F1 was heated to 40 °C and centrifugated at 3000 rpm for 30 min to remove α-lactalbumin precipitate. The supernatant was then filtered through Whatman No. 4 filter paper to remove the residual α-lactalbumin. Then, the filtrate (F2) was collected and the pH was adjusted to 6.0 by adding NH_4_OH drop by drop. (NH_4_)_2_SO_4_ (47.6 g/100 mL) was then added to F2 to precipitate goat milk β-lactoglobulin and was filtered by Whatman No. 1 filter paper (GE Healthcare, Boston, MA, USA). Finally, the solids were dissolved in 30 mL pure water and dialyzed (12~14 kDa) in pure water for 48 h with water changed every 2 h. After dialysis, the β-lactoglobulin was freeze-dried.

### 3.3. Analysis of β-Lactoglobulin by Reversed Phase High Performance Liquid Chromatography (RP-HPLC)

Purity of the isolated β-lactoglobulin was measured by reversed phase high performance liquid chromatography (RP-HPLC, UltiMate 3000, ThermoFisher Scientific, Waltham, MA, USA) equipped with DAD UV–Vis absorption detector [34]. The operation conditions are listed below: injection volume of 10 μL, detection wavelength of 214 nm, and flow rate of 1.0 mL/min. The mobile phase of A and B consisted of water containing 0.1% trifluoroacetic acid (TFA) and acetonitrile containing 0.1% TFA, respectively. The solvent gradient program was 80A:20B to 50A:50B from 0 to 20 min, and then 50A:50B to 80A:20B from 20 to 30 min. Standard β-lactoglobulin with the concentration of 0.1–10 mg/mL was used to plot the standard curve. An equation (y = 100.64 x + 0.9921) with regression coefficient of 0.9987 were obtained. Based on the linear regression equation, the purity of the β-lactoglobulin from raw goat milk was calculated to be 88.50 ± 2.50%.

### 3.4. High Intensity Ultrasound (HIU) Treatment on Goat Milk β-Lactoglobulin

The powdered β-lactoglobulin was dissolved in 10 mM phosphate buffer (PB, pH = 6.5) to make a final concentration at 3 mg/mL [26]. Sodium azide (0.4 mg/mL, w/v) was added as an antibacterial agent. Thirty milliliters of the β-lactoglobulin solution was treated by an Ultrasonic Processor (JY92-IIDN, Ningbo Scientz, Zhejiang, China) equipped with a 6-mm high grade titanium alloy probe (amplitude, 220 μm). Samples were treated by HIU (650 W, 20 kHz) at different amplitudes (20%, 30%, and 40%) and times (0, 1, 3, 5, 7, and 10 min). Temperature was controlled using an ice-water bath at 25~30 °C. After ultrasound treatment, the sample was divided into two parts. One part was freeze-dried and the other one was stored at 4 °C for further analysis. The untreated β-lactoglobulin solution was used as the control.

### 3.5. Surface Hydrophobicity Measurements

Surface hydrophobicity (H_0_) of samples were analyzed according to the method of Chen et al. with some modifications [25]. Briefly, samples (3 mg/mL) were diluted with phosphate buffer (PB 10 mM, pH 6.5) to concentrations of 0.125, 0.25, 0.5, and 1 mg/mL. Fifty microliters of 8-anilino-1-naphtalensulfonic acid (ANS, 0.008 M) were added to 4 mL sample solution. After vortex, the mixture was incubated for 15 min in the dark. The fluorescence intensity (FI) of the protein-ANS conjugates was measured by a fluorescence spectrophotometer (RF-6000, Shimadzu, Japan) at 365 nm (excitation, slit 5 nm) and 484 nm (emission, slit 5 nm) with a scanning speed of 10 nm/s. The index of the protein hydrophobicity was expressed as the initial slope of FI versus protein concentration (calculated by linear regression analysis). Each experiment was carried out for three trials and each trial was conducted in triplicate.

### 3.6. Intrinsic Fluorescence Spectra

Intrinsic fluorescence of the protein samples was analyzed by a fluorescence spectrophotometer (RF-6000, Shimadzu, Japan). Samples (3 mg/mL) were diluted with PB (10 mM, pH 6.5) to obtain the protein concentration at 0.25 mg/mL. All solutions were excited at 295 nm (slit = 5 nm), and emission spectra were recorded from 300 to 450 nm (slit = 5 nm) with a scanning speed of 200 nm/min.

### 3.7. Fourier Transform Infrared Spectra (FTIR)

FTIR spectra of the protein samples was analyzed using a Nicolet 6700 FTIR spectrometer equipped with attenuated total reflectance (ATR) ZnSe crystal (ThermoFisher Scientific, Waltham, MA, USA) and Omnic software (Version 9.2, ThermoFisher Scientific, Waltham, MA, USA). Freeze-dried samples were ground in an agate mortar before measurement. Each spectra were the average of 32 scans at 4 cm^−1^ resolution. Measurements were recorded between 4000 and 550 cm^−1^. Amide I band from 1600 to 1700 cm^−1^ was analyzed with Fourier transform deconvolution and second derivative peak fitting using Peak Fit software (version 4.12, SeaSolve Software Inc, San Jose, CA, USA). The corresponding relationships between the absorption peak and secondary structure were as follows: 1650~1660 cm^−1^ for α-helices; 1610~1637 cm^−1^ for β-sheets; 1660~1695 cm^−1^ for β-turns; and 1637~1650 cm^−1^ for random coils.

### 3.8. Native and Sodium Dodecyl Sulfate (SDS) Polyacrylamide Gel Electrophoresis (PAGE) of Treated and Untreated Samples

Protein profiles were analyzed using SDS-PAGE (reducing and non-reducing) and native-PAGE [35]. For reducing conditions, the sample (80 µL) was mixed with 5× sample loading buffer (20 µL) P0015 (Beyotime Biotechnology, Shanghai, China) and then boiled for 3 min. For non-reducing conditions, a portion of diluted sample (40 µL) was mixed with 5× non-reducing loading buffer (10 µL) CW0028S (Cwbiotechnology, Taizhou, Jiangsu, China) and then boiled for 3 min. The SDS-PAGE was conducted with 12% separating gel and 5% stacking gel, which were prepared by a SDS-PAGE Gel Preparation Kit P0012 (Beyotime Biotechnology, Shanghai, China). Running buffer (1 L) was made with Tris glycine SDS buffer solution P0552 (Beyotime Biotechnology, Shanghai, China) in the ratio of 1:20 with MilliQ water. Electrophoresis was conducted by a Mini-protean Tetra Electrophoresis System (Bio-Rad, Hercules, CA, USA). The voltage was increased from 80 to 120 V. After electrophoresis, gels were stained with Coomassie blue fast staining solution P0017 (Beyotime Biotechnology, Shanghai, China) for approximately 30 min and then de-stained with deionized water. Protein ladder ranging from 10 to 170 kDa (ThermoFisher Scientific, Waltham, MA, USA) was used as molecular weight standard.

For native-PAGE, samples were mixed with sample buffer P0016N (Beyotime Biotechnology, Shanghai, China) without boiling. Protein profile were run in gels formed by 5% and 12% acrylamide gels (stacking and separating, respectively) without adding SDS. Running buffer was made with Tris glycine buffer solution P0556 (Beyotime Biotechnology, Shanghai, China) in the ratio of 1:20 with MilliQ water.

### 3.9. Differential Scanning Calorimetry (DSC)

Thermal analysis of the protein samples was conducted with differential scanning calorimetry (DSC 3, Mettler Toledo, Zurich, Switzerland) supported with the pre-installed thermal analysis software (star-e). An empty pan of equal weight was used as a reference. Approximately 5~6 mg of solid sample was weighed and added into the aluminum pans. The temperature was maintained at 20 °C for 10 min and then ramped from 20 °C to 120 °C at a heating rate of 5 °C/min. Nitrogen (50 mL/min) was used as cooling gas.

### 3.10. Transmission Electron Microscopy (TEM)

Microstructure of goat milk β-lactoglobulin samples treated by HIU (20%, 30%, and 40% amplitude, 10 min) and the untreated control were examined by TEM. The sample (3 mg/mL, 100 μL) was dropped on a carbon film supported by a Cu grid. Filter paper was used to absorb the excess sample after 40 min. The sample, after staining with uranyl acetate (20 mg/mL) solution for 5 min, was dried at ambient temperature and then underwent microstructure observation with an H-7800 transmission electron microscopy in high contrast imaging mode (Hitachi High-Technologies, Tokyo, Japan) operated at an acceleration voltage of 100 kV.

### 3.11. Statistical Analysis

All the data were presented as mean ± standard deviation (SD). SPSS Version 24 (SPSS Inc. Chicago, IL, USA) were used to analyze the significant differences of data among the samples. Homogeneity between data was checked using Levene’s test. After a homogeneity check, one-way analysis of variance (ANOVA) and least squared differences (LSD) model were used for the analysis of homogeneous data while Dunnett’s test was used for heterogeneous data. Significant level between samples were set at *p* = 0.05. All the figures were drawn by Origin 2020 (OriginLab Corp., Northampton, MA, USA). 

## 4. Conclusions

Effects of high intensity ultrasound on physiochemical and structural properties of goat milk β-lactoglobulin were studied. No major change was observed in the molecular weight of protein by electrophoretic patterns, but high-intensity ultrasound induced changes in the secondary and tertiary structures of goat milk β-lactoglobulin. High intensity ultrasound improved the thermal stability of the β-lactoglobulin molecules. Larger aggregated protein was found in high intensity ultrasound treated samples than untreated samples. Ultrasound treatment would destroy the internal hydrophobic interactions of protein molecules, unfolding of the protein molecular and accelerating protein molecular motion, which resulted in protein aggregation. The results of this study provide helpful information about the impact of ultrasound treatment on goat milk β-lactoglobulin.

## Figures and Tables

**Figure 1 molecules-25-03637-f001:**
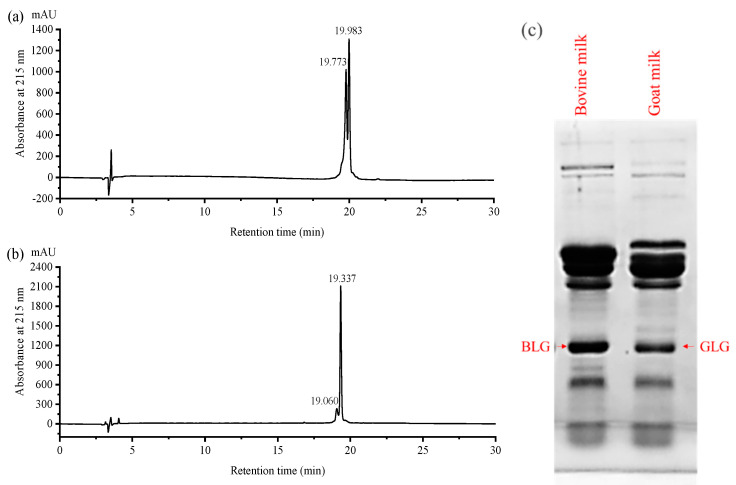
Reversed Phase High Performance Liquid Chromatography (RP-HPLC) chromatogram of bovine milk β-lactoglobulins (**a**) and goat milk β-lactoglobulins (**b**), and Sodium Dodecyl Sulfate Polyacrylamide Gel Electrophoresis (SDS-PAGE) of bovine and goat milk (**c**). Bands of bovine and goat milk β-lactoglobulins in SDS-PAGE pattern named Bovine milk β-Lactoglobulin (BLG) and goat milk β-Lactoglobulin (GLG), respectively.

**Figure 2 molecules-25-03637-f002:**
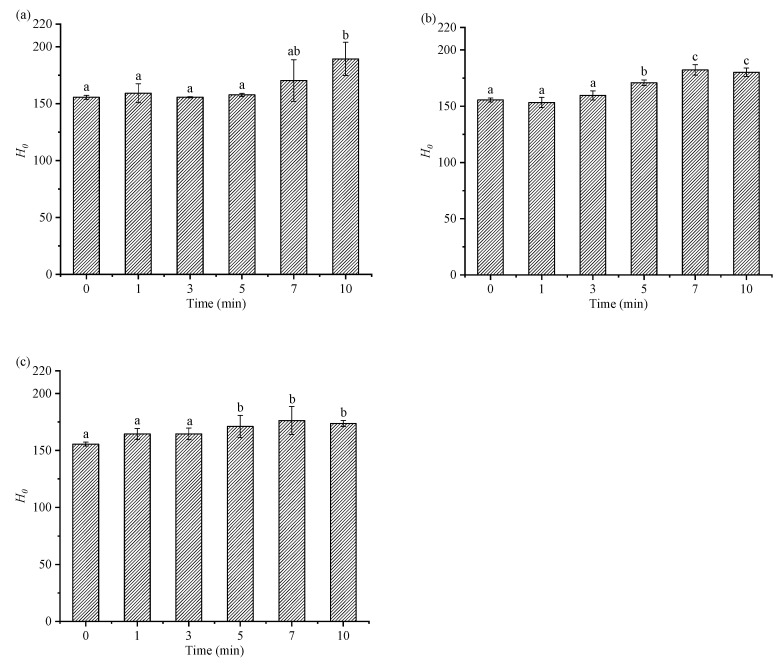
Surface hydrophobicity (H_0_, ×100,000) of goat milk β-lactoglobulin samples treated at different amplitudes (20% (**a**), 30% (**b**), 40% (**c**)) and different treat times (0, 1, 3, 5, 7, 10 min). Completely different lowercase letter indicates significant differences at *p* < 0.05.

**Figure 3 molecules-25-03637-f003:**
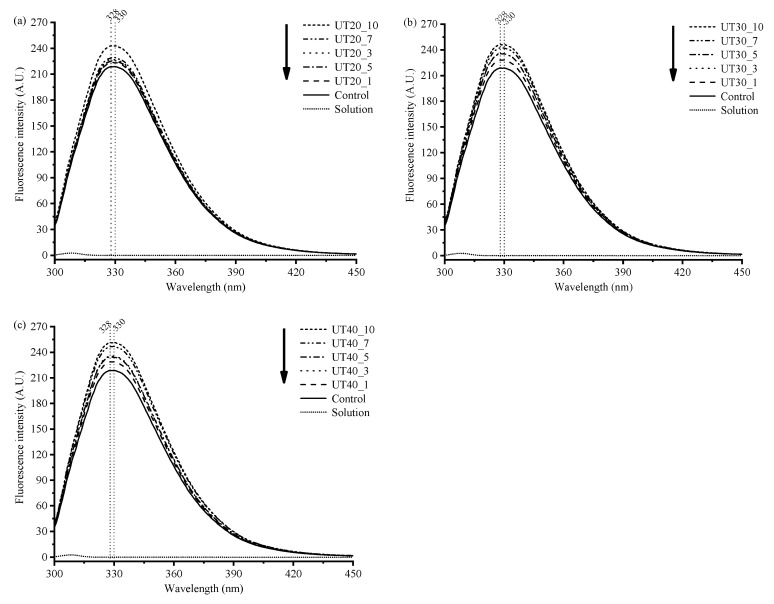
Intrinsic fluorescence spectra of goat milk β-lactoglobulin samples treated at different amplitudes (20% (**a**), 30% (**b**), 40% (**c**)) and different treat times (0, 1, 3, 5, 7, 10 min). Phosphate buffer (PB) solution was set as background.

**Figure 4 molecules-25-03637-f004:**
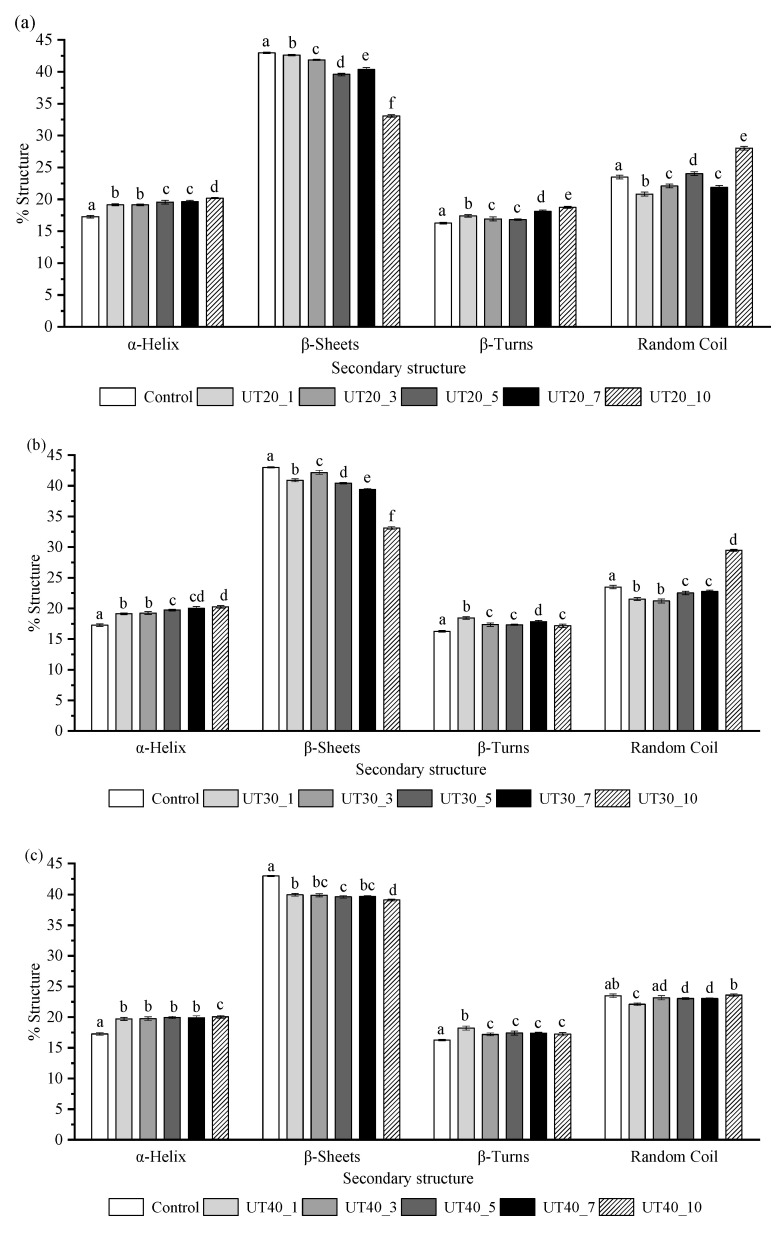
Secondary structure content of goat milk β-lactoglobulin samples treated at different amplitudes 20% (**a**), 30% (**b**), 40% (**c**) and different treat times (0, 1, 3, 5, 7, 10 min). Note: Completely different lowercase letter in the same HIU amplitude and secondary structure indicates significant difference at *p* < 0.05.

**Figure 5 molecules-25-03637-f005:**
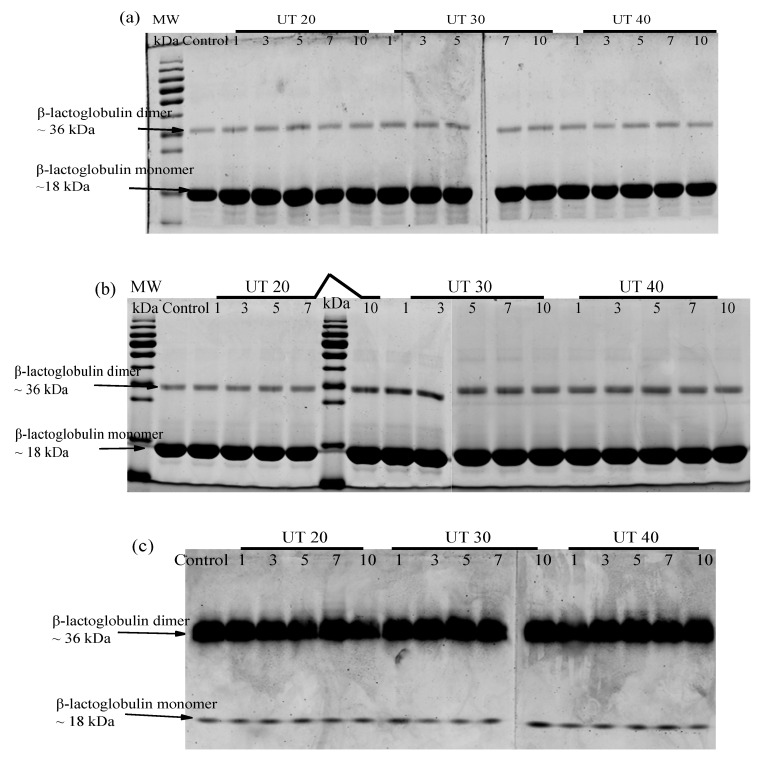
Effects of ultrasound treatment on molecular weight. (**a**) Reducing Sodium Dodecyl Sulfate (SDS) - Polyacrylamide Gel Electrophoresis (PAGE), (**b**) Non-reducing SDS-PAGE, (**c**) Native PAGE of goat milk β-lactoglobulin in different amplitudes (20%, 30%, 40%) and different treatment times (0, 1, 3, 5, 7, 10 min).

**Figure 6 molecules-25-03637-f006:**
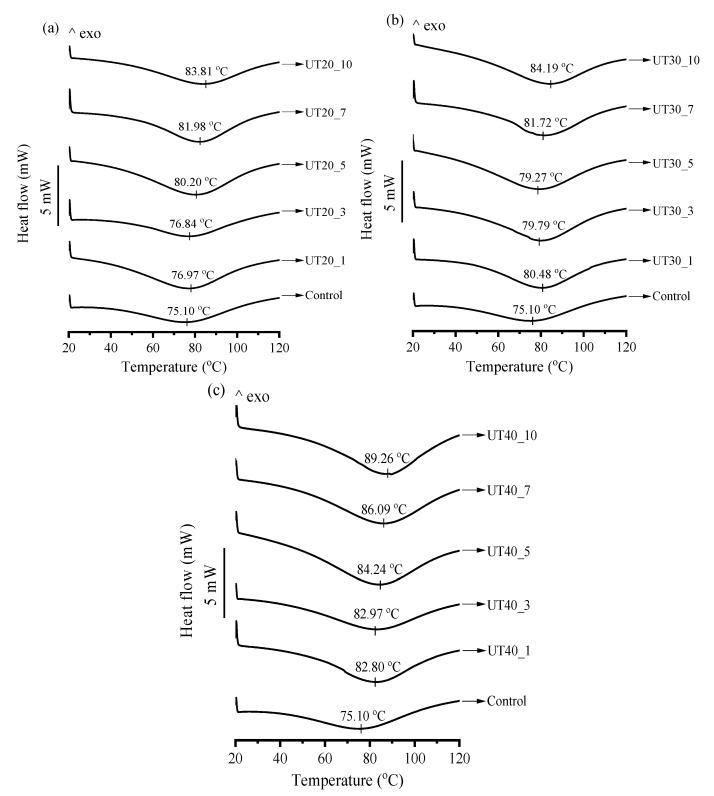
Effects of ultrasound treatment on Differential Scanning Calorimetry (DSC) curves of goat milk β-lactoglobulin samples treated at different amplitudes (20% (**a**), 30% (**b**), 40% (**c**)) and different treat times (0, 1, 3, 5, 7, 10 min).

**Figure 7 molecules-25-03637-f007:**
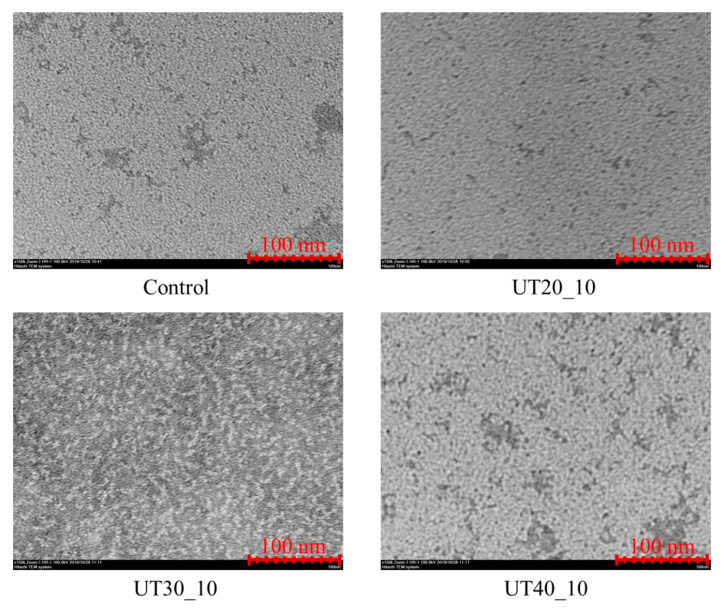
Transmission electron microscopy photographs (×150,000) of the goat milk β-lactoglobulin treated by HIU in different amplitudes (20%, 30%, and 40%) for 10 min.

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
