# Peer review of "Effects of High Intensity Ultrasound on Physiochemical and Structural Properties of Goat Milk β-Lactoglobulin"

_molecules, 2020, doi:10.3390/molecules25163637_

Round 1
Reviewer 1 Report
The authors of this manuscript present effects of high intensity ultrasound (HIU) on biophysical properties of goat milk beta-lactoglobulin (LG). They used various techniques that allowed them to reveal physicochemical and structural changes in goat LG. As a result, the authors showed that HIU treatment induced changes in biophysical properties of goat LG leading to structural rearrangement. These studies provide new insight into protein properties and food components after HIU treatment. The manuscript is written consistently and logically.Detailed suggestions to improve the quality of the manuscript are added below:
1. In my opinion this paper would gain quality if the authors compare results for goat LG with bovine LG (experimental work is required). Authors referred to the literature data many times, but I suggest the authors prepare one experiment for bovine LG to check its properties at described conditions. 2. Material and Method Section: lanes: 105-109 and lanes: 159-162 - the text is incorrectly formatted3. Results and discussion Section: Fig.1. the red line that indicates LGs should be removed from the picture- it covers the band. Please modify resolution of the figure 1c (especially description- lane numbers, "c"- should be the same as on the Fig.1a,b.4. Please explain the presence of two peaks on RP HPLC chromatogram of bovine LG5. Table 1- It will be beneficial if authors compare the surface hydrophobicity index of goat LF with bovine LG.6. Table 1- can the authors explain the difference in surface hydrophobicity for samples treated for 7 and 10 min (20 % amplitude)? This difference is not visible for 30 and 40 % amplitude. 7. Table 1/ I suggest show these results in the form of graphs.8. Figure 2- Authors described the effect of HIU on intrinsic fluorescence of goat LG. They found that the emission maximum was shifted from 328 t0 330 nm. This effect is not visible on the graphs presented in the fig. 2. Similarly, a red shift to 280 nm is not visible for samples treated with HIU, presented in Fig. 3.9. Figure 4- lowercase/uppercase description is illegible and difficult to understand. The figure legend is unclear. In my opinion the authors have to explain what it means: aA, bB, bdB, adC... etc. It is totally incomprehensible and needs detailed clarification.10. Figure 5 a-c - molecular weight of standard should be indicated more clearly.The authors do not need to indicate all molecular weight. It will be enough if the authors indicate molecular weights corresponding to monomer and dimer of goat LG. Fig. 5C- Mw is not marked on this figure. Molecular mass of the bands should be indicated. 11. Figure 7 - the scale is not clearly visible. Please change the position of the scale bar.
Author Response
Dear Dr. Qi,
Thank you for the valuable comments from you and the Reviewers on our manuscript titled: “Effects of high intensity ultrasound on physiochemical and structural properties of goat milk β-lactoglobulin” (molecules-849420). The manuscript has been revised according to the comments from the Reviewers. The following are the responses to the comments for your reference:
Responses to Reviewer 1:
Q 1: In my opinion this paper would gain quality if the authors compare results for goat LG with bovine LG (experimental work is required). Authors referred to the literature data many times, but I suggest the authors prepare one experiment for bovine LG to check its properties at described conditions.
Authors: Thank you for your suggestion. It is a very good point. Bovine milk beta-LG has been widely studied. In this study, we mainly aimed to investigate the effects of HIU applied at various amplitudes and durations on the physiochemical and structural properties of goat milk β-lactoglobulin. Perhaps, we will do a comparison work as suggested in the future.
Q 2: Material and Method Section: lanes: 105-109 and lanes: 159-162 - the text is incorrectly formatted
Authors: We apologize for the error. Please see the revised manuscript in Page 3, Lines 128-132 and Page 5, Lines 183-187.
Q 3: Results and discussion Section: Fig.1. the red line that indicates LGs should be removed from the picture- it covers the band. Please modify resolution of the figure 1c (especially description- lane numbers, "c"- should be the same as on the Fig.1a, b.
Authors: Thank you for your helpful comment. We have carefully revised it. Please see Figure 1 (Page 6, Line 226) in the revised manuscript.
Q 4: Please explain the presence of two peaks on RP HPLC chromatogram of bovine LG
Authors: Thank you for your comment. Two peaks on RP-HPLC chromatogram of bovine milk β-lactoglobulin represent bovine β-lactoglobulin A and B variants, respectively. Please see the revised manuscript in Page 5, Lines 217-219.
Page 5, Lines 217-219: The retention time of main peaks for goat milk β-lactoglobulin was were at 19.060 and 19.337 min (Figure 1b) while that those for bovine milk β-lactoglobulin was were 20.019. and 19.983 min (represent bovine milk β-lactoglobulin A and B variants Figure 1a).
Q 5: Table 1- It will be beneficial if authors compare the surface hydrophobicity index of goat LG with bovine LG.
Authors: Thank you for your suggestion. The aim of this study was to compare the changes in the structure of goat milk β-lactoglobulin during HIU treatment in different amplitudes and durations. The comparation about hydrophobicity index for goat and bovine milk β-lactoglobulin will be carried out in our next study. The data will be reported in due course.
Q 6: Table 1- can the authors explain the difference in surface hydrophobicity for samples treated for 7 and 10 min (20 % amplitude)? This difference is not visible for 30 and 40 % amplitude.
Authors: Thank you for your comment. There were no significant differences between samples treated by HIU for 7 and 10 min at 20, 30, and 40% amplitude.
Q 7: Table 1/ I suggest show these results in the form of graphs.
Authors: Thank you for your comment. Table 1 has been shown in graph (Figure 2). Please see the revised manuscript in Page 7, Lines 244-245.
Q 8: Figure 2- Authors described the effect of HIU on intrinsic fluorescence of goat LG. They found that the emission maximum was shifted from 328 t0 330 nm. This effect is not visible on the graphs presented in the fig. 2. Similarly, a red shift to 280 nm is not visible for samples treated with HIU, presented in Fig. 3.
Authors: Thank you for your comment. Emission maximum has been labeled. Please see the revised manuscript in Page 8-9, Lines 276-277 (Figure 3).
Q 9: Figure 4- lowercase/uppercase description is illegible and difficult to understand. The figure legend is unclear. In my opinion the authors have to explain what it means: aA, bB, bdB, adC... etc. It is totally incomprehensible and needs detailed clarification.
Authors: Thank you for your suggestion. The description of lowercase letter was carefully revised, please see the manuscript in Page 12, Lines 322-324.
Page 12, Lines 322-324: Note: Completely different lowercase letter in the same HIU amplitude and secondary structure indicates significant difference at p < 0.05.
Q 10: Figure 5 a-c - molecular weight of standard should be indicated more clearly. The authors do not need to indicate all molecular weight. It will be enough if the authors indicate molecular weights corresponding to monomer and dimer of goat LG. Fig. 5C- Mw is not marked on this figure. Molecular mass of the bands should be indicated.
Authors: Thank you for your comment. The molecular weights of goat milk β-lactoglobulin monomer and dimer were shown clearly in Figure 5. Please see the revised manuscript in Page 13, Lines 349-351.
Q 11: 11. Figure 7 - the scale is not clearly visible. Please change the position of the scale bar.
Authors: Thank you for your suggestion. The scale of TEM was carefully modified. Please see the revised manuscript in Page 17, Lines 389.
Reviewer 2 Report
The authors investigated the effects of the high intensity ultrasound (HIU) treatment on physicochemical and structural properties of goat beta-lactoglobulin (bLG). They compared these properties of HIU-treated goat bLG sample with those of untreated sample using various methods, including RP-HPLC, ANS and intrinsic fluorescence, UV and FTIR spectra, SDS- and native-PAGE, DSC, and TEM. The main effects of the HIU treatment were increase in the ANS fluorescence and decrease in the beta structure contents, indicating that the HIU treatment induces the partial collapse of the calyx structure and subsequent exposure of some hydrophobic residues. However, the DSC results showed that the heat-resistance of goat bLG was enhanced by the HIU treatment.
Although these data are novel and contain some information, the reviewer would like to raise two significant concerns on the present manuscript. One is, it is ambiguous what kind of HIU-induced (macroscopic) phenomena the authors wanted to investigate or supposed. As the authors mentioned in Introduction, the HIU treatment might be applicable to modification of material properties and food processing. The author also quoted the HIU effect on bovine milk to change the immunoreactivity (it is known that bovine bLG can cause human allergy). On the other hand, are there any known HIU-induced phenomena of goat milk or goat bLG solution? It is better to mention clearly what kind of HIU-induced phenomena on goat bLG is targeted by this research.
Another concern is that there is little discussion about the effect of HIU on the goat bLG from a molecular viewpoint. As mentioned above, the main results were enhanced ANS fluorescence, beta-structure reduction, and enhanced thermal stability. However, the author did not present any supposed structural change of the goat bLG molecule, which can explain all of the obtained results. The reviewer also supposed a protein-protein interaction via exposed hydrophobic residues. However, the author did not perform a confirmation experiment of this hypothesis. As a result, the authors have just enumerated the observation results and have linked these observations neither to explanations of the macroscopic change of goat bLG solution (or goat milk) nor to deductions of structural changes of goat bLG molecule. This journal is “Molecules”, so the latter concern might be a critical point.
Therefore, the reviewer judged that the present manuscript is not acceptable for publication. In addition, the reviewer listed several points to be modified.
- Line 45: “amino acids at” should be “amino acid substitutions at”.
- Line 84: After the addition of Na2SO4 (this step seems salting-in), supernatant or precipitation, which fraction was taken for the next step? Please describe apparently.
- Line 117: Is it required to prepare samples with various protein concentration? Is it not enough with only one sample to measure ANS fluorescence?
- Line 137. Why did the author measure FTIR on the freeze-dried sample? The reviewer supposes that, in any cases, the protein sample, which will be subject to the HIU treatment, must be in wet or solution state.
- 1 (a) and (b): Why do bovine and goat bLGs show the multiple peaks? What kind of heterogeneity was included?
- Line 208: The definition of H0 should be explained. As this quantity is unit-less, unfamiliar reader cannot imagine how to calculate or obtain this value.
- Table 1: It is better to present these results in a dot-and-line graph. The information significance of differences might not be necessary here.
- Line 226: “exit” might be “emit”.
- Line 227: “exited” might be “excited”.
- Line 228: “at exit wavelength” might be “at excitation wavelength”.
- Line 233: “a polarity environment which could change” might be “a polar environment which resulted from a change”.
- Line 236: The author mentioned that “, thus leading to an increase in the fluorescence intensity” However, an exposure of Trp side chain to the aqueous environment generally results in a decrease in its fluorescence. The authors’ interpretation is opposite to the general understanding. Please add additional explanation to this interpretation.
- 3: Nowadays, difference of absorption spectra is not usually used to assess conformational change because the sensitivity of this method is not so high and instead there are many better methods. Indeed, in this study, the spectral changes were not so significant and did not give obvious information. Therefore, it seems better to remove these results from the manuscript.
- Line 260: “quantitative” might be “quantify”.
- Line 264: What is “The interior unfolding process”? The reviewer cannot understand this terminology.
- Line 272: The authors mentioned “the cavitation phenomenon induced by ultrasonic treatment could expose some of the hydrophobic regions”. What is the mechanism of the exposure of the hydrophobic residues by ultrasonic treatment? Please introduce a plausible explanation.
- 4: It is better to show these data in dot-and-line presentation with a horizontal axis of time than the current bar presentation. Then, the individual data (a-Helix, b-sheet, b-turn, and random coil) under the same amplitudes of HIU treatment should be shown in one panel. The information significance of differences might not be necessary here, too.
- Line 289: What does the “light” mean?
- Line 302: “remarkable” should be “remarkably”.
- 5(a): Why were there still dimer band (~36 kDa) even under reducing conditions?
- 5 (d)~(f): It is better to expand the sample elution region (retention time of 17~23 minutes of the horizontal axes) and remove other regions. It is unnecessary to include null regions in the figure.
- Line 317: Temperature is not a quantity. Thus, the expression “increased the Tpeak” is not appropriate. “raised” is better.
- Line 322: “Vander Waals” should be “van der Waals interactions”.
- Line 318: The author mentioned “the thermal stability of the protein was improved by HIU treatment”. However, it is ambiguous the stability of what kind of structure the authors are mentioning; for example, unfolding of the native structure or dissociation of intermolecular association? The author should clearly mention what kind of structural change is thought to be attributed to the observed endothermic peak.
- 6: Why did the author measure DSC on the freeze-dried sample? It is the same question as comment 4 and related to the above comment. Especially, it is considered that the hydrated water plays an important role for determination of the stability of the protein native structure. Because hydrated water is absent or significantly low in the freeze-dried samples, the stability of the protein molecule will be significantly different with those in the aqueous solution.
- Line 319: The author mentioned “TEM images showed aggregated after HIU treatment”. However, the reviewer cannot distinguish the image from those of the control sample. Clearly mention how the author concluded so.
- 7: Please include scale bars in each panel.
Author Response
Dear Dr. Qi,
Thank you for the valuable comments from you and the Reviewers on our manuscript titled: “Effects of high intensity ultrasound on physiochemical and structural properties of goat milk β-lactoglobulin” (molecules-849420). The manuscript has been revised according to the comments from the Reviewers. The following are the responses to the comments for your reference:
Responses to Reviewer 2:
Q 1: Line 45: “amino acids at” should be “amino acid substitutions at”.
Authors: Thank you for your comment. New words “amino acid substitutions at” have been added. Please see the revised manuscript in Page 2, Line 58.
Q 2: Line 84: After the addition of Na2SO4 (this step seems salting-in), supernatant or precipitation, which fraction was taken for the next step? Please describe apparently.
Authors: Thank you for the helpful comment. The precipitate was discarded, and the supernatant (Filtrate, F1) was taken for the next step. Please see the revised manuscript in Page 3, Line 108.
Page 3, Line 108: The precipitate was discarded.
Q 3: Line 117: Is it required to prepare samples with various protein concentration? Is it not enough with only one sample to measure ANS fluorescence?
Authors: Thank you for your suggestion. According to reference we cited (Chen and Others, 2019), calculating the surface hydrophobicity (H0), at least four concentrations of each sample were required.
Chen, W.; Wang, W.; Ma, X.; Lv, R.; Balaso Watharkar, R.; Ding, T.; Ye, X.; Liu, D. Effect of pH-shifting treatment on structural and functional properties of whey protein isolate and its interaction with (-)-epigallocatechin-3-gallate. Food Chem. 2019, 274, 234-241.
Q 4: Line 137. Why did the author measure FTIR on the freeze-dried sample? The reviewer supposes that, in any cases, the protein sample, which will be subject to the HIU treatment, must be in wet or solution state.
Authors: Thank you for your suggestion. The absorption peak of H2O was at 1640 cm-1. It could interfere with the peak fitting procedure of Amide I band (1700-1600 cm-1).
Q 5: (a) and (b): Why do bovine and goat bLGs show the multiple peaks? What kind of heterogeneity was included?
Authors: Thank you for your comment. Two peaks eluting on RP-HPLC chromatogram of bovine milk β-lactoglobulin at 19.773 and 19.983 min represent bovine β-lactoglobulin A and B variants, respectively. However, to our knowledge, very little information about heterogeneity for goat milk β-lactoglobulin was published. We will consider your suggestions for one of our ongoing projects.
Q 6: The definition of H0 should be explained. As this quantity is unit-less, unfamiliar reader cannot imagine how to calculate or obtain this value.
Authors: Thank you for your comment. Surface hydrophobicity (H0) of protein represents the index of the number of hydrophobic groups present on the surface of protein molecules. The counting process has been described in the revised manuscript (Page 4, Lines 141-146).
Page 4, Lines 141-146: The fluorescence intensity (FI) of the protein-ANS conjugates was measured by a Fluorescence Spectrophotometer (RF-6000, Shimadzu, Japan) at 365 nm (excitation, slit 5 nm) and 484 nm (emission, slit 5 nm) with a scanning speed of 10 nm/s. The index of the protein hydrophobicity was expressed as the initial slope of FI versus protein concentration (calculated by linear regression analysis).
Q 7: Table 1: It is better to present these results in a dot-and-line graph. The information significance of differences might not be necessary here.
Authors: Thank you for your suggestion. The new histogram was used to represent the data in Table 1. Please see Page 7, Lines 244-245 in the revised manuscript.
Page 7, Lines 244-245: Figure 2. Surface hydrophobicity (H0, × 100,000) of goat milk β-lactoglobulin samples treated at different amplitudes (20% a, 30% b, 40% c) and different treat times (0, 1, 3, 5, 7, 10 min). Completely different lowercase letter indicates significant differences at p < 0.05.
Q 8: Line 226: “exit” might be “emit”.
Authors: We apologize for the typo. Please see Page 8, Line 260 in the revised manuscript.
Q 9: Line 227: “exited” might be “excited”.
Authors: We apologize for the error. Please see Page 8, Line 261 in the revised manuscript.
Q 10: Line 228: “at exit wavelength” might be “at excitation wavelength”.
Authors: We apologize for the error. Please see Page 8, Line 262 in the revised manuscript.
Q 11: Line 233: “a polarity environment which could change” might be “a polar environment which resulted from a change”.
Authors: Thank you for your valuable comments. A new sentence has been shown in Page 8, Lines 266-268 in the revised manuscript.
Page 8, Lines 266-268: The red shift of fluorescence emission wavelength suggested that more fluorescent amino acid was exposed into a polarity environment which resulted in the change of conformation of protein.
Q 12: Line 236: The author mentioned that “, thus leading to an increase in the fluorescence intensity” However, an exposure of Trp side chain to the aqueous environment generally results in a decrease in its fluorescence. The authors’ interpretation is opposite to the general understanding. Please add additional explanation to this interpretation.
Authors: Thank you for your valuable comments. The increment of FI might be that some Trp groups initially inside the molecules were exposed to the surface after ultrasound treatment. In addition, partial unfolding of protein molecular resulted by the ultrasound treatment might decrease the internal quenching, which contributed to the increase of FI. The additional explanation has been added to the revised manuscript in Page 8, Lines 271-273.
Page 8, Lines 271-273: And partial unfolding of protein molecular resulted by the HIU treatment might decrease the internal quenching, which contributed to the increase of fluorescence intensity [21].
Q 13: Nowadays, difference of absorption spectra is not usually used to assess conformational change because the sensitivity of this method is not so high and instead there are many better methods. Indeed, in this study, the spectral changes were not so significant and did not give obvious information. Therefore, it seems better to remove these results from the manuscript.
Authors: We agreed with your comment and section about UV absorption has been deleted.
Q 14: Line 260: “quantitative” might be “quantify”.
Authors: We apologize for the error. Please see Page 10, Line 299 in the revised manuscript.
Q 15: Line 264: What is “The interior unfolding process”? The reviewer cannot understand this terminology.
Authors: Thank you for your helpful comments. “The interior unfolding process” means the unfolding of β-lactoglobulin molecules, which was attributed to the acoustic cavitation during HIU treatment.
Q 16: Line 272: The authors mentioned “the cavitation phenomenon induced by ultrasonic treatment could expose some of the hydrophobic regions”. What is the mechanism of the exposure of the hydrophobic residues by ultrasonic treatment? Please introduce a plausible explanation.
Authors: Thank you so much for your valuable comment. Ultrasound-induced protein modification is often attributed to acoustic cavitation. Cavitation-induced activities, such as high shear by micro and macro-streaming, shock waves, and water jets, alter the molecular structure of protein. The mechanism of the exposure of the hydrophobic residues by ultrasonic treatment, mainly due to higher shear forces, resulted in breakdown of polypeptides chain, thus, disruption of intramolecular disulphide linkage and hydrophobic interactions.
Q 17: It is better to show these data in dot-and-line presentation with a horizontal axis of time than the current bar presentation. Then, the individual data (a-Helix, b-sheet, b-turn, and random coil) under the same amplitudes of HIU treatment should be shown in one panel. The information significance of differences might not be necessary here, too.
Authors: Thank you for your comment. Figure 4 has been modified according to your suggestion and another reviewer. Please see Page 11, Lines 315-317 in the revised manuscript.
Q 18: Line 289: What does the “light” mean?
Authors: We apologize for the error. “Light” was changed as “increased”. Please see Page 13, Line 333 in the revised manuscript.
Q 19: Line 302: “remarkable” should be “remarkably”.
Authors: We apologize for the error. According to the suggestion from other reviewers, section (RP-HPLC analysis) of the word mentioned above has been deleted.
Q 20: 5(a): Why were there still dimer band (~36 kDa) even under reducing conditions?
Authors: Thank you for your comment. Content of sample loading buffer (20 µL) used may be not enough to dissociate the dimer of β-lactoglobulin. However, this phenomenon would not influence the conclusion of this study. Similar results were observed by previous study listed below:
(1) Rahaman T, Vasiljevic T, Ramchandran L. Conformational changes of β-lactoglobulin induced by shear, heat, and pH-Effects on antigenicity. J. Dairy Sci, 2015, 98(7):4255-4265.
(2) Park, H.W. Kim, D.Y. Shin, W.S. Fucoidan improves the structural integrity and the molecular stability of beta- lactoglobulin. Food Sci. Biotechnol. 2018, 27, 1247-1255.
Q 21: Figure 5 (d)~(f): It is better to expand the sample elution region (retention time of 17~23 minutes of the horizontal axes) and remove other regions. It is unnecessary to include null regions in the figure.
Authors: Thank you for your comment. According to the suggestion from other reviewers, section of RP-HPLC analysis (Figure 5d ~f) has been deleted.
Q 22: Line 317: Temperature is not a quantity. Thus, the expression “increased the Tpeak” is not appropriate. “raised” is better.
Authors: Thank you for your comment. “increased” has been changed as “raised”. Please see Page 15, Line 364 in the revised manuscript.
Q 23: Line 322: “Vander Waals” should be “van der Waals interactions”.
Authors: We apologize for the typo. Please see Page 17, Lines 387-388 in the revised manuscript.
Q 24: Line 318: The author mentioned “the thermal stability of the protein was improved by HIU treatment”. However, it is ambiguous the stability of what kind of structure the authors are mentioning; for example, unfolding of the native structure or dissociation of intermolecular association? The author should clearly mention what kind of structural change is thought to be attributed to the observed endothermic peak.
Authors: Thank you for your valuable comment. HIU treatment unfolding of the native structure of goat milk β-lactoglobulin. The endothermic peak may be attributed to the destruction of intramolecular bonds in protein molecular. Please see the specific information in Page 15, Lines 367-370.
Page 15, Lines 367-370: HIU treatment induced a certain degree of molecular unfolding of the goat milk β-lactoglobulin which disorder the stable compact fold of native goat milk β-lactoglobulin. More energy may be required to denatured the unfolding structure by dissociation of intramolecular bonds such as covalent bonds.
Q 25: Why did the author measure DSC on the freeze-dried sample? It is the same question as comment 4 and related to the above comment. Especially, it is considered that the hydrated water plays an important role for determination of the stability of the protein native structure. Because hydrated water is absent or significantly low in the freeze-dried samples, the stability of the protein molecule will be significantly different with those in the aqueous solution.
Authors: Thank you for your valuable comment. The process of gasification of water could interfere with the identification of values of Tpeak for β-lactoglobulin samples in DSC test.
Q 26: Line 319: The author mentioned “TEM images showed aggregated after HIU treatment”. However, the reviewer cannot distinguish the image from those of the control sample. Clearly mention how the author concluded so.
Authors: Thank you for your comment. The white areas represented molecules of goat milk β-lactoglobulin in TEM images. Native goat milk β-lactoglobulin (Control) has a spherical shape with a diameter of about 3.6 nm. After HIU treatment, samples showed larger irregular microscopic particle clusters (> 10 nm) than control. Detailed information was added in Page 17, Lines 383-386.
Page 17, Lines 383-386: For TEM images (Figure 7), the white areas represented molecules of goat milk β-lactoglobulin. Native goat milk β-lactoglobulin (Control) has a spherical shape with a diameter of about 3.6 nm. After HIU treatment, samples showed larger irregular microscopic particle clusters (> 10 nm) than control.
Q 27: Fig.7: Please include scale bars in each panel.
Authors: Thank you for your suggestion. The scale of TEM was carefully modified. Please see the revised manuscript in Page 17, Lines 389.
Reviewer 3 Report
Main comment:
The article provides new and interesting information, but does not emphasize the purpose of the research carried out, to which the results could be applied, or whether it is simply the assimilation and application of certain methods. In this case, The Abstract and Introduction sections need additional information. In the conclusions, the trends obtained could be described in more detail. The conclusions could be supplemented by a review of the possible application of the results obtained (see Abstract).
Comments:
lines 159-169. Please use regular font size for this text, not italic.
Line 180: insert state and country for SPSS.
Fig. 5. I suggest the figures related to SDS-Page to present separately from the RP-HPLC chromatograms or remove the latter figures, because practically there are the same results presented.
Also, the gel wells should be removed from the SDS-Page pictures.
The manuscript needs additional English style revision.
Author Response
Dear Dr. Qi,
Thank you for the valuable comments from you and the Reviewers on our manuscript titled: “Effects of high intensity ultrasound on physiochemical and structural properties of goat milk β-lactoglobulin” (molecules-849420). The manuscript has been revised according to the comments from the Reviewers. The following are the responses to the comments for your reference:
Responses to Reviewer 3:
Q 1: The Abstract and Introduction sections need additional information. In the conclusions, the trends obtained could be described in more detail.
Authors: Thank you for your comment. Sections of abstract, introduction and conclusions were revised carefully. Please see the specific information in Page 1, Lines 17-30, Page 2, Lines 71-73, Lines 75-83, Page 18, Lines 394-396, Lines 399-402.
Page 1, Lines 17-30: This study aimed to compare the effects of high intensity ultrasound (HIU) applied at various amplitudes (20 ~ 40 %) and for different durations (1 ~ 10 min) on physiochemical and structural properties of goat milk β-lactoglobulin. No significant change was observed in the protein electrophoretic patterns by sodium dodecyl sulphate polyacrylamide gel electrophoresis (SDS-PAGE). Deconvolution and second derivative of the FTIR spectra showed that percentage of β-sheet of goat milk β-lactoglobulin was significantly decreased while those of α-helix and random coils increased after HIU treatment The surface hydrophobicity index and intrinsic fluorescence intensity of samples was enhanced and increased with increasing HIU amplitude or time. DSC results exhibited that HIU treatments improved the thermal stability of goat milk β-lactoglobulin. Transmission electron microscopy (TEM) of samples showed that goat milk β-lactoglobulin microstructure had changed and it contained larger aggregates when compared with the untreated goat milk β-lactoglobulin sample. Data suggested that HIU treatments resulted in secondary and tertiary structural changes of goat milk β-lactoglobulin and improved its thermal stability.
Page 2, Lines 71-73, The application of non-thermal techniques has potential to modify the functional properties of proteins thus; recently this technique has become great area of interest for researchers.
Page 2, Lines 75-83: Certain studies investigated the changes in molecular structure after HIU treatment, which induced alterations in free sulfhydryl groups, particle sizes, surface hydrophobicity, and secondary structures of proteins. Several authors speculated that the impact of ultrasound on the chemical and physical changes of protein, mainly due to higher shear forces, resulted in breakdown of polypeptides chain, thus, disruption of intermolecular disulphide and non-covalent hydrophobic interactions [20, 27, 28]. HIU can modify the secondary and tertiary structures of proteins, which lead to the changes of the functional properties such as emulsifying activity, gelling properties and foaming capacity of proteins [13-16].
Page 18, Lines 394-396: No major change was observed in the molecular weight of protein by electrophoretic patterns, but high-intensity ultrasound induced changes in the secondary and tertiary structures of goat milk β-lactoglobulin.
Page 18, Lines 399-402 Larger aggregated protein was found in high intensity ultrasound treated samples than untreated samples. Ultrasound treatment would destroy the internal hydrophobic interactions of protein molecules, unfolding of the protein molecular and accelerating protein molecular motion, which resulted in protein aggregation.
Q 2: lines 159-162. Please use regular font size for this text, not italic.
Authors: We apologize for the error. Format has been revised carefully. Please see the revised manuscript in Page 5, Lines 183-187.
Q 3: Line 180: insert state and country for SPSS.
Authors: Thank you for your comment. State and country of SPSS has been added in Page 5, Line 205.
Q 4: Fig. 5. I suggest the figures related to SDS-Page to present separately from the RP-HPLC chromatograms or remove the latter figures, because practically there are the same results presented.
Authors: We agree with your suggestion. Results of RP-HPLC from “3.5. Effects of HIU treatment on molecular weight of goat milk β-lactoglobulin” has been deleted in the revised manuscript.
Q 5: Also, the gel wells should be removed from the SDS-Page pictures.
Authors: Thank you for your comment. The gel wells from SDS-PAGE pictures were removed. Please see the revised manuscript in Page 13, Line 349-351.
Q 6: The manuscript needs additional English style revision.
Authors: Thank you for your suggestion. The English of this manuscript has been carefully edited.
Reviewer 4 Report
In the method section 2.9 the authors should change the style. Lines 159-162
The authors should check the identity of amino acid sequences for both proteins.
Line 270, Please explain what does mean “partial collapse of calyx”
Line 320 TEM images (Figure 7) showed aggregates in goat milk β-lactoglobulin after HIU treatment. The authors should explain what UT20_10, UT30_10, UT40_10 means in Figure 7. What concentration of protein was used for these images. More descriptions are needed for this figure. It may be necessary to demonstrate what happens to β-lactoglobulin from bovine's milk after treatment with HIU using a TEM image.
Author Response
Dear Dr. Qi,
Thank you for the valuable comments from you and the Reviewers on our manuscript titled: “Effects of high intensity ultrasound on physiochemical and structural properties of goat milk β-lactoglobulin” (molecules-849420). The manuscript has been revised according to the comments from the Reviewers. The following are the responses to the comments for your reference:
Responses to Reviewer 4:
Q 1: In the method section 2.9 the authors should change the style. Lines 159-162
Authors: We apologize for the error. Format has been revised carefully. Please see the revised manuscript in Page 5, Lines 183-187.
Q 2: The authors should check the identity of amino acid sequences for both proteins.
Authors: Thank you for your valuable comment. Actually, eight different amino acids were found from the sequences between bovine (PDB 3BLG) and goat (PDB 4OMX) milk β-lactoglobulin. The mistake has been revised. Please see Page 2, Lines 58-61 in the revised manuscript.
Page 2, Lines 58-61: However, there are eight amino acids substitutions at N-terminal Leu, Asp 53, Asp 64, Val 118, Asp 130, Ser 150, Glu 158 and Ile 162 in bovine milk β-lactoglobulin (Database accession number, PDB 3BLG) compared with N-terminal Ile, Asn 53, Gly 64, Ala 118, Lys 130, Ala 150, Gly 158 and Val 162 in goat milk β-lactoglobulin(PDB 4OMX).
Q 3: Line 270, Please explain what does mean “partial collapse of calyx”
Authors: Thank you for your comment. Eight of the β-sheet (strands A-H) fold up into a flattened β-barrel that is also called a calyx in native β-lactoglobulin. Reduction of β-sheet in goat milk β-lactoglobulin after HIU treatment may involve partial collapse of calyx.
Q 4: Line 320 TEM images (Figure 7) showed aggregates in goat milk β-lactoglobulin after HIU treatment. The authors should explain what UT20_10, UT30_10, UT40_10 means in Figure 7. What concentration of protein was used for these images. More descriptions are needed for this figure. It may be necessary to demonstrate what happens to β-lactoglobulin from bovine's milk after treatment with HIU using a TEM image.
Authors: Thank you for your comment. UT20_10, UT30_10 and UT40_10 in Figure 7 means β-lactoglobulin treated by HIU in amplitudes 20, 30, 40% for 10 min (Page 18, Line 391). The concentration of protein used for these images was 3 mg/mL (Page 5, Line 197). The additional descriptions about results from TEM were added, please see the revised manuscript. Please see Page 17, Lines 383-386.
Page 17, Lines 383-386: For TEM images (Figure 7), the white areas represented molecules of goat milk β-lactoglobulin. Native goat milk β-lactoglobulin (Control) has a spherical shape with a diameter of about 3.6 nm. After HIU treatment, samples showed larger irregular microscopic particle clusters (> 10 nm) than control.
Reviewer 5 Report
To the author
The paper is of good scientific quality but some minor modification could be done:
Line 90 pag.3….”to participate goat milk”….I think is…..”to precipitate goat milk”
Line 385 pag.14 …”17.17Stanic-Vucinic”…to delete the second 17
Line 432/433 pag.15… “36. Note. Column or rows with different lowercase or uppercase indicates significant differences at p< 0.05 respectively”…..to delete reference 36. I think in the References it is not the right place.
I suggest a deeper conclusion.
Author Response
Dear Dr. Qi,
Thank you for the valuable comments from you and the Reviewers on our manuscript titled: “Effects of high intensity ultrasound on physiochemical and structural properties of goat milk β-lactoglobulin” (molecules-849420). The manuscript has been revised according to the comments from the Reviewers. The following are the responses to the comments for your reference:
Responses to Reviewer 5:
Q 1: Line 90 pag.3….”to participate goat milk”….I think is…..”to precipitate goat milk”
Authors: We apologize for the typo. Please see the revised manuscript in Page 3, Line 113.
Q 2: Line 385 pag.14 …”17.17Stanic-Vucinic”…to delete the second 17
Authors: We apologize for the error. Please see the revised manuscript in Page 19, Line 450.
Q 3: Line 432/433 pag.15… “36. Note. Column or rows with different lowercase or uppercase indicates significant differences at p< 0.05 respectively”…..to delete reference 36. I think in the References it is not the right place.
Authors: We apologize for the error. Format has been revised. Please see the revised manuscript.
Q 4: I suggest a deeper conclusion.
Authors: Thank you for your comment. A new deeper conclusion has been added to the revised manuscript in Page 18, Lines 393-404.
Page 18, Lines 393-404: Effects of high-intensity ultrasound on physiochemical and structural properties of goat milk β-lactoglobulin were studied. No major change was observed in the molecular weight of protein by electrophoretic patterns, but high-intensity ultrasound induced changes in the secondary and tertiary structures of goat milk β-lactoglobulin. High intensity ultrasound improved the thermal stability of the β-lactoglobulin molecules. Larger aggregated protein was found in high intensity ultrasound treated samples than untreated samples. Ultrasound treatment would destroy the internal hydrophobic interactions of protein molecules, unfolding of the protein molecular and accelerating protein molecular motion, which resulted in protein aggregation. The results of this study provide helpful information about the impact of ultrasound treatment on goat milk β-lactoglobulin.
Round 2
Reviewer 1 Report
The authors provided a comprehensive response to the requested information. These answers are satisfied for me, and for this reason I recommend this manuscript for publication in its present form.
Reviewer 2 Report
Because the reviewer feels that the revised manuscript still have a number of concerns, the judgement "reject" was not changed.
There are three major concerns left in the revised manuscript.
A. One major concern is that the authors do not clearly present or suggest a molecular mechanism of the HIU treatment to protein molecules yet. Probably, the authors suppose that the HIU treatment induced partial unfolding of the GLG native structure, which subsequently induces intermolecular aggregation. The resultant aggregation was heat-resistant and the heat resistance can be improved according to the length and amplitude of the HIU treatment. The author should explain all of the experimental data along with these story.
B. Another major concern is the usage of the freeze-dried sample for the FT-IR and DSC measurements. Although the authors answered the reason of the usage of the freeze-dried sample, these were not scientific reasons. If the authors want to remove the water band, the author can use heavy water. If the authors want to detect thermal stability of the aggregate in solution, the author can monitor the thermal denaturation by using FT-IR or other spectroscopic methods. Anyway, because the stability of protein sample in freeze-dried state should be significantly different from those in solution state, the usage of the freeze-dried sample cannot be justified.
C. The third one is the word in the title "physiochemical". Is it OK?
And, a number of authors' responses to the original comments were not satisfactory. Please reconsider the following comments (the number is the same as the original list).
4. (related to the major concern B) The reviewer supposes that, in any cases, the protein sample, which will be subject to the HIU treatment, must be in wet or solution state. The reviewer request re-experiments in solution state using heavy water buffer or other methods.
7. New Figure 2: It is better to present these results in a dot-and-line graph. The information significance of differences might not be necessary here.
15. What is “The interior unfolding process”? Generally, this word does not refer to "the unfolding of β-lactoglobulin molecules".
17. New Figure 4: It is better to show these data in dot-and-line presentation with a horizontal axis of time than the current bar presentation. Then, the individual data (a-Helix, b-sheet, b-turn, and random coil) under the same amplitudes of HIU treatment should be shown in one panel (with a common vertical axis representing time). The information significance of differences might not be necessary here, too.
24. (related to the major concern A) The author added the following sentences in line 301; "HIU treatment induced a certain degree of molecular unfolding of the goat milk β-lactoglobulin which disorder the stable compact fold of native goat milk β-lactoglobulin. More energy may be required to denatured the unfolding structure by dissociation of intramolecular bonds such as covalent bonds." However, it is still ambiguous what mechanism the authors are supposing. One reason is inappropriate usage of words. Another reason is the ambiguity of "the covalent bonds". What kind of "covalent bond" do the author supposed? The third is that the sentences above seems inconsistent with New Figure 5c; why were not there larger species than dimer?
25. (related to the major concern B) Why did the author measure DSC on the freeze-dried sample? It is the same question as comment 4 and related to the above comment. Especially, it is considered that the hydrated water plays an important role for determination of the stability of the protein native structure. Because hydrated water is absent or significantly low in the freeze-dried samples, the stability of the protein molecule will be significantly different with those in the aqueous solution. The reviewer also request re-experiments in solution state to monitor the thermal stability using other methods.
26. The author added the following sentence; "Native goat milk β-lactoglobulin (Control) has a spherical shape with a diameter of about 3.6 nm. After HIU treatment, samples showed larger irregular microscopic particle clusters (> 10 nm) than control." Then, please expand the TEM images and clearly indicate where the spherical aggregate of 3.6 nm diameter and microscopic (macroscopic?) particle cluster of > 10 nm in diameter (or length?) are.